# A novel diagnostic approach for assessing pulmonary blood flow distribution using conventional X-ray angiography

**Takuya Sakaguchi**[1], **Yuichiro Watanabe**[1], **Masashi Hirose**[1], **Kohta Takei**[2], **Satoshi Yasukochi**[2] *

1 Canon Medical Systems Corporation, Otawara, Tochigi, Japan, 2 Heart Center, Nagano Children's Hospital, Azumino, Nagano, Japan

* maplesyjuha@me.com

**Data Availability Statement:** All relevant data are within the manuscript and its Supporting Information files.

## Abstract

### Objective

Quantitative assessment of pulmonary blood flow distribution is important when determining the clinical indications for treating pulmonary arterial branch stenosis. Lung perfusion scintigraphy is currently the gold standard for quantitative blood flow measurement. However, it is expensive, cannot provide a real-time assessment, requires additional sedation, and exposes the patient to ionizing radiation. The aim of this study was to investigate the feasibility of a novel technology for measuring pulmonary blood flow distribution in each lung by conventional X-ray pulmonary angiography and to compare its performance to that of lung perfusion scintigraphy.

### Methods

Contrast-enhanced X-ray pulmonary angiography images were acquired at a frame rate of 30 frames per second. The baseline mask image, obtained before contrast agent injection, was subtracted from subsequent, consecutive images. The time-signal intensity curves of two regions of interest, established at each lung field, were obtained on a frame-to-frame basis. The net increase in signal intensity within each region at the torrent period during the second cardiac cycle before contrast agent enhancement over the total lung field was measured, and the right-to-left ratio of the signal intensity was calculated. The right-to-left ratio obtained with this approach was compared to that obtained with scintigraphy. Agreement of the right-to-left ratio between X-ray angiography and lung scintigraphy measurements was assessed using linear fitting with the Pearson correlation coefficient.

### Result

The calculation of the right-to-left ratio of pulmonary blood flow by our kinetic model was feasible for seven children as a pilot study. The right-to-left ratio of pulmonary blood flow distribution calculated from pulmonary angiography was in good agreement with that of lung

**Funding:** We also would like to note that this research was funded by Canon Medical Systems Corporation. The funder provided support in the form of salaries for authors TS, YW, and MH, but did not have any additional role in the study design, data collection and analysis, decision to publish, or preparation of the manuscript. The specific roles of these authors are articulated in the 'author contributions' section.

**Competing interests:** TS, YW, MH were employees of Cannon Medical Systems Corporation. KT and SY were pediatric cardiologist of Nagano Children's Hospital. The authors did not receive any compensation including honorary from Cannon Medical Systems Corporation. This does not alter our adherence to PLOS ONE policies on sharing data and materials.

perfusion scintigraphy, with a Pearson correlation coefficient of 0.91 and a slope of linear fit of 1.2 (p<0.005).

## Conclusion

The novel diagnostic technology using X-ray pulmonary angiography from our kinetic model can feasibly quantify the right-to-left ratio of pulmonary blood flow distribution. This technology may serve as a substitute for lung perfusion scintigraphy, which is quite beneficial for small children susceptible to radiation exposure.

## Introduction

Quantitative assessment of the right-to-left ratio of pulmonary blood flow distribution is important when determining the clinical indications for treating pulmonary arterial branch stenosis, which is often found in cases with congenital heart disease such as tetralogy of Fallot and transposition of the great artery before and after surgical repair [1]. Lung perfusion scintigraphy (LS) is the current gold standard for quantifying pulmonary blood flow and determining indications for intervention either by catheter treatment or by surgery [2, 3]. The current standard of care for patients with an unbalanced pulmonary blood flow distribution due to pulmonary branch stenosis requires several time-consuming diagnostic steps to make a decision, including both pulmonary angiography and lung scintigraphy on different days. Then, patients needed to return to the catheterization laboratory to undergo balloon angioplasty or stent implantation. This treatment approach is a significant burden to both the patient and the health care providers. If one can assess the right-to-left ratio of the pulmonary blood flow immediately after angiography, patients can be treated in the same session. A key to success in achieving this goal is determining whether the right-to-left ratio of pulmonary blood flow distribution can be measured only by conventional X-ray pulmonary angiography (XA) consistent with LS. For this reason, we created a novel technology that can quantitatively assess the right-to-left ratio of pulmonary blood flow distribution in the clinical setting.

## Theory and algorithm

Many studies have measured blood flow distribution in humans with mathematical tracer kinetic models [4–8]. In this kinetic model using tracer concentration, a two-compartment model is adopted, as shown in (1):

$$\frac{dC_p(t)}{dt} = K_1 C_a(t) - k_2 C_p(t) \tag{1}$$

where $C_a(t)$ is the tracer concentration in the artery (g/ml) and $C_p(t)$ is the tracer concentration in the organ tissue region (g/g). $K_1$ is the transfer constant of the tracer from the artery to the tissue region (ml/min/g). $k_2$ is the transfer constant of the tracer from the tissue region to the vein (1/min). We applied this compartment models to calculate the tracer kinetics of the lung region using LS and XA, as shown in **Fig 1**.

### A. Tracer kinetic model for scintigraphy

The tracers used in lung perfusion scintigraphy are radioisotope particles (also called radiopharmaceuticals) whose diameters range from 10–30 μm. These radiopaque particles, injected via veins, become trapped in the pulmonary capillary bed (whose vessel diameters range from

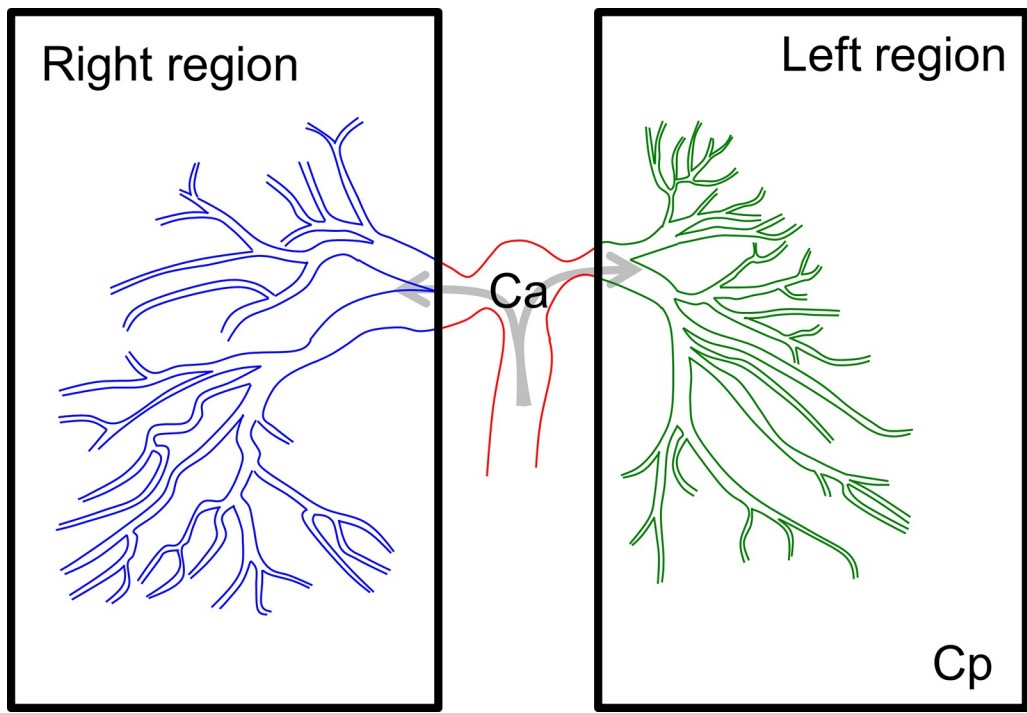

**Fig 1. ROI establishment for the right and left lung regions.** The region of interest (ROI) was set to cover the entire field of the right and left lung separately. Ca represents the tracer concentration in the main pulmonary trunk, and Cp represents the tracer concentration in each lung region.

6–10 μm). The radiation from these trapped particles is measured by a scintigraphy imaging system [9]. Since all particles are trapped, there is no leakage from tissue to vessels, and the $k_2 C_p$ term in (1) can be neglected. Hence, after integration of (1), the signal intensity measured by the system camera, $S_{LS}$, can be described by (2) [10]:

$$S_{LS}(t) = k_{LS}C_{p,RI}(t) = k_{LS}K_1 \int_0^t C_{a,RI}(\tau) \cdot e^{-\frac{\tau}{T_d}} d\tau \tag{2}$$

where $C_{a,RI}(t)$ is the tracer concentration in the pulmonary trunk (g/ml) at a given time (t) and $C_{p,RI}(t)$ is the tracer concentration in the lung region (g/g). $K_1$ is the transfer constant of tracer (ml/min/g). $k_{LS}$ is the calibration factor of the scintigraphy camera (count/g of tracer), and $T_d$ is the tracer decay time constant. With this approach, the measured signal count is converted to the number of tracers to quantify blood flow. The time signal intensity curve of this model, $S_{LS}(t)$, is shown in **Fig 2(A)**.

To calculate the right-to-left ratio with scintigraphy, the lung is separated into two regions: the right region and left region (as demonstrated), and hence, the ratio is determined as follows:

$$S_{LS}^{Right}(T) : S_{LS}^{Left}(T) = k_{LS}C_{p,RI}^{Right}(t) : k_{LS}C_{p,RI}^{Left}(t) = K_1^{Right} : K_1^{Left} \tag{3}$$

Therefore, the above equation indicates that the right-to-left ratio obtained by scintigraphy is equivalent to the ratio of the tracer transfer constants, $K_1^{Right} : K_1^{Left}$.

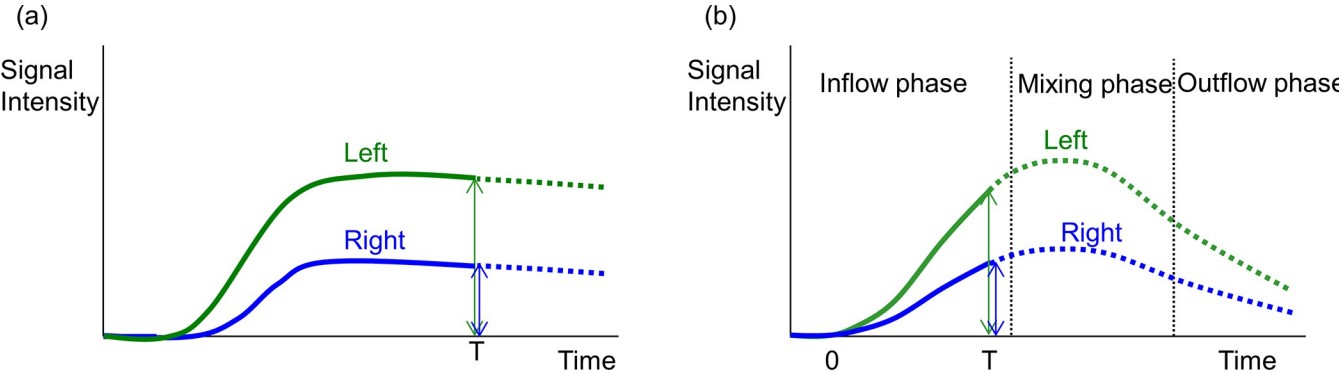

**Fig 2. Kinetic model of the signal intensity of agents in each modality.** The signal intensity curve for radioactive particles trapped in the right and left pulmonary bed separately by scintigraphy is shown in (a). The ratio of the signal intensity in each lung is maintained over time. However, the signal intensity curve for the contrast agent in each lung as measured by angiography shows time-dependent changes that depend on the pulmonary circulation, and thus the right-to-left lung distribution ratio also changes (b). Therefore, the measurement time window for pulmonary blood distribution should be set to before the contrast agent passes through the pulmonary capillary bed to the left atrium.

## B. Tracer kinetic model for angiography

Conventional pulmonary X-ray angiography uses an iodine contrast agent as a tracer. The injected contrast agent passes through the main pulmonary arterial trunk to both the right and left pulmonary artery branches, then goes through the pulmonary capillaries to reach the pulmonary vein, and finally arrives at the left atrium. Kinetic model (1) can be integrated and expressed as [11–13]:

$$C_{p,Iodine}(t) = K_1 \int_0^t C_{a,Iodine}(\tau)d\tau - k_2 \int_0^t C_{p,Iodine}(\tau)d\tau \tag{4}$$

where $C_{a,Iodine}(t)$ is the contrast concentration in the pulmonary trunk (g/ml) and $C_{p,Iodine}(t)$ is the contrast concentration in the lung region (g/g). $K_1$ is the transfer constant of the contrast agent from the artery to the lung region (ml/min/g). $k_2$ is the transfer constant of the contrast agent from the lung region to the pulmonary veins (1/min).

The time intensity curve of XA is a bell-shaped curve, as shown in **Fig 2(B)**, which is different from that of LS, shown in **Fig 2(A)**. The cause of this difference is attributed to the fact that iodine contrast agent passes through the capillary bed to reach the pulmonary vein without trapping, while the radioisotope particles are trapped and stay in the capillary bed. Because of this discrepancy, a simple comparison of the time-intensity curves between modalities shows that they are quite different. To overcome this discrepancy, the kinetic model is modified and adjusted to match the blood flow kinetics by setting the optimal time window for XA to obtain the equivalent calculated value for LS. The inflow phase is determined as the time interval that starts just before injection of the contrast agent in the main pulmonary arterial trunk and ends just before contrast agent flows out from the lung region to the pulmonary vein (**Fig 2(B)**). In XA, if the time window for calculating the signal intensity of the contrast agent is limited to the inflow phase, the outflow from the capillaries to the pulmonary vein can be ignored. Hence, (4) can be written as:

$$C_{p,Iodine}(t) = K_1 \int_0^t C_{a,Iodine}(\tau)d\tau \tag{5}$$

A baseline image of X-ray angiography I(0) is obtained at time t = 0 prior to contrast injection, and a consecutive images I(t) are obtained at time t after contrast injection. The

absorption relation can be expressed as (6) [14]. The log-subtraction signal intensity $S_{XA}(t)$, which is also known as digital subtraction angiography (DSA) [15], is determined as (7):

$$I(t) = I(0) \times e^{-\mu L} \tag{6}$$

$$S_{XA}(t) = \ln\left(\frac{I(0)}{I(t)}\right) = \mu \times L \tag{7}$$

where $\mu$ is the X-ray linear attenuation coefficient (1/cm) and L is the X-ray path length in the contrast material (cm). $\mu$ can be rewritten as the product of two values $\tau$ and d (defined below); then, the above equation can be expressed as:

$$S_{XA}(t) = \tau \times d \times L = \tau \times C_{Iodine}(t) \times r \times L \tag{8}$$

where $\tau$ is the mass attenuation coefficient (cm$^2$/g), d is the density of iodine (g/cm$^3$), $C_{Iodine}$ is the concentration of contrast agent (g/g), and r is the mixing ratio of iodine and water in contrast agent (g/cm$^3$). By combining with (5), (8) can be rewritten as:

$$S_{XA}(t) = \tau \times r \times L \times K_1 \int_0^t C_{a,Iodine}(\tau)d\tau \tag{9}$$

Therefore, the right-to-left ratio of pulmonary blood flow obtained using X-ray angiography can be calculated from (9) and written as:

$$S_{XA}^{Right} : S_{XA}^{Left} = \tau r L \times K_1^{Right} \int_0^t C_{a,Iodine}(\tau)d\tau : \tau r L \times K_1^{Left} \int_0^t C_{a,Iodine}(\tau)d\tau = K_1^{Right} : K_1^{Left} \tag{10}$$

## C. Right-to-left flow distribution ratio

In lung scintigraphy, the right-to-left ratio of pulmonary blood flow distribution, $S_{LS}^{Right} : S_{LS}^{Left}$, is proportional to the ratio of the tracer transfer constant $K_1^{Right} : K_1^{Left}$, as shown in (3). As shown in (10), $K_1^{Right} : K_1^{Left}$ is proportional to the XA signal intensity ratio, $S_{XA}^{Right} : S_{XA}^{Left}$. Therefore, the ratio of signal intensity in the inflow phase of XA for the right and left lung regions is equivalent to the ratio of the signal intensity in LS.

## D. Setting a large region of interest to cover the entire lung

To obtain an accurate ratio of pulmonary blood flow with LS, the measurement must cover the entire lung field of each lung, eliminating the heart and middle mediastinum field, as demonstrated in **Fig 3**. In XA, the region of interest should be set in the same manner to measure the concentration of iodine contrast agent. The inhomogeneity of the signal intensity in each pixel of XA may be observed in the clinical setting but can be canceled out if one sets a sufficiently large ROI. Once the contrast agent flows into the ROI, the transit delay in the ROI does not affect the measurement. The subtle motion and deformation of the lung by respiration and patient motion can also be canceled out. Catheter motion along with cardiac contraction can also be canceled out both by subtraction and by setting a sufficiently large ROI. Consequently, the calculated result from this novel quantification of pulmonary angiography could be equivalent to that of LS.

## E. Setting the time window for calculation at the torrent period

The time window for calculating the signal intensity of the ROI in XA should be limited to the inflow phase, as described in the previous section. The inflow phase corresponds to the time

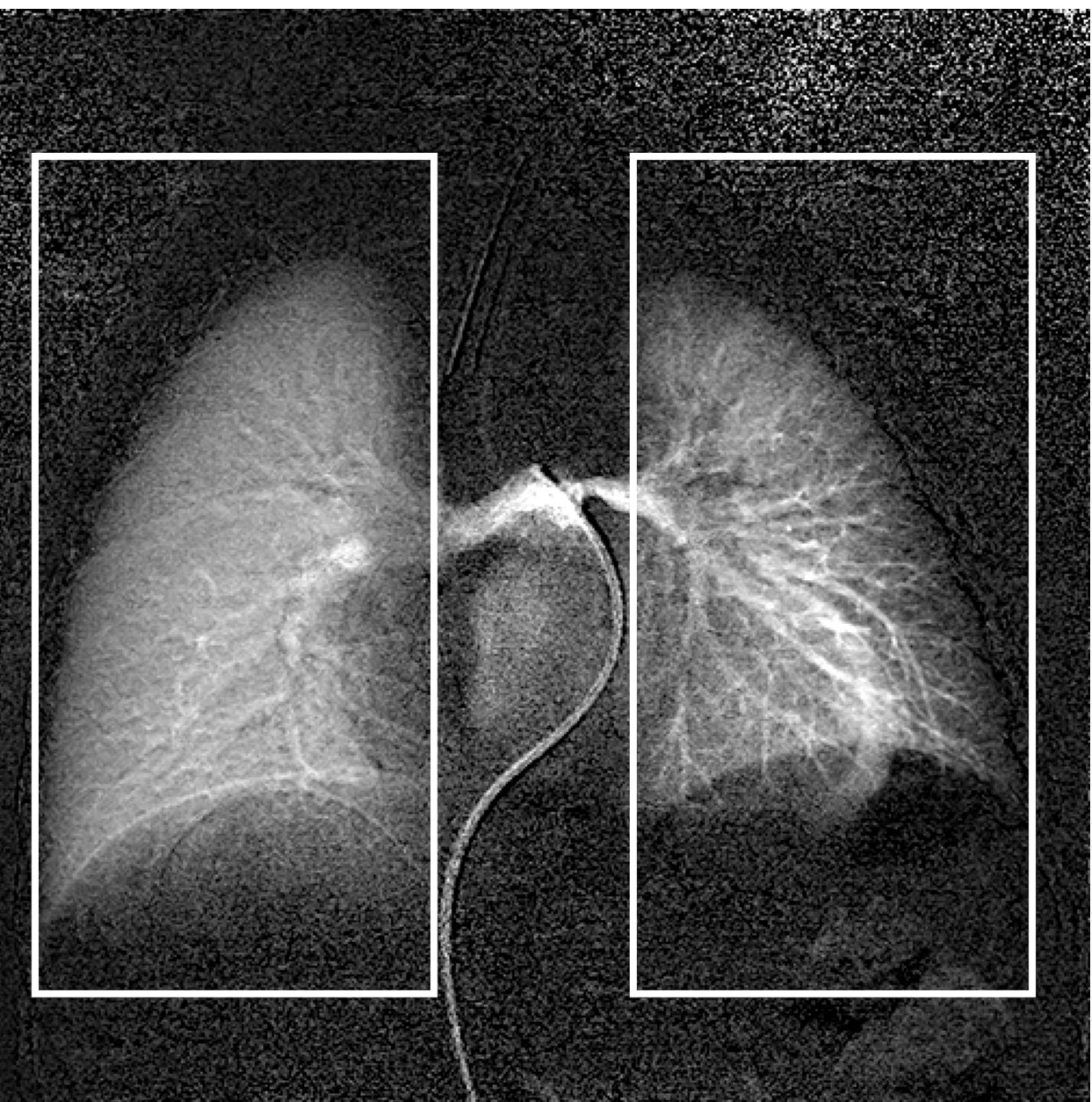

**Fig 3. ROI establishment for measurement for each lung.** Each lung field should be covered when data acquisition for calculating pulmonary blood distribution is performed by angiography as described in the text.

interval that starts just before injection of the contrast agent and ends just before outflow of the contrast agent from the lung region to the pulmonary vein. It takes five cardiac cycles for the blood in the right ventricle to pass through the pulmonary circulation to the left atrium [16]. Since the inflow phase can be assumed to be half of the pulmonary circulation, only the first 2.5 cardiac cycles after the contrast agent flows into the main pulmonary arterial trunk can be used for data acquisition to calculate the signal intensity.

A typical time-signal intensity curve in the ROI of both lung regions demonstrates a staircase-like shape and significant discrepancy in the first cardiac cycle after contrast injection

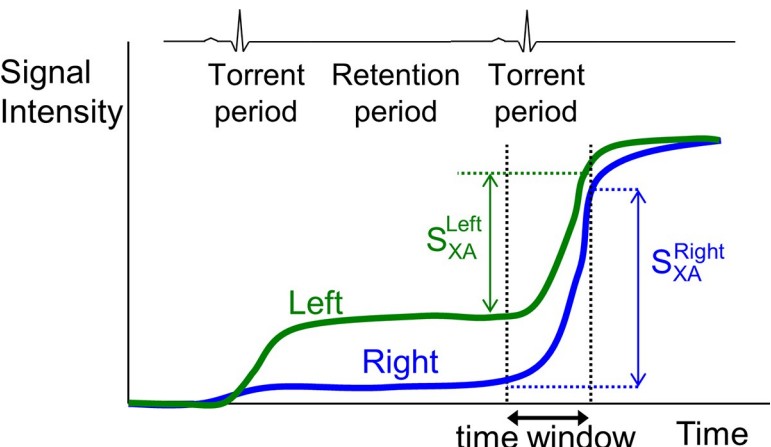

**Fig 4. Time-signal intensity curves of contrast agent in each lung field.** The time-intensity curve of contrast agent in each lung field over a cardiac cycle is shown. Since the baseline signal intensity of contrast in each lung is different, the net increase in signal intensity of each lung ($S_{XA}^{Left}$, $S_{XA}^{Right}$) over a torrent period during the second cardiac cycle after injection is used for calculating the pulmonary blood flow distribution for analysis.

(**Fig 4**). This large variation in time signal intensity is due to the poor mixing of contrast agent with various blood flow streams into the pulmonary trunk (**Fig 5**). A typical example is shown in **Fig 6,** where contrast agent flows into the left lung during the first cardiac cycle after contrast injection, which makes measurement impossible. To eliminate this variance in contrast agent concentration after injection, the time window is set in the second cardiac cycle.

In addition, the time window is set during the torrent time period, which represents the period during which the contrast agent is torrentially discharged from the pulmonary arteries to the capillary bed. For practical purposes, it is defined as approximately 10% of the R-R interval. In this time window, the slope of the time-signal intensity curve is much steeper than that at the other periods (**Fig 4**). The right-to-left ratio of pulmonary blood flow can be more accurately assessed in the torrent period.

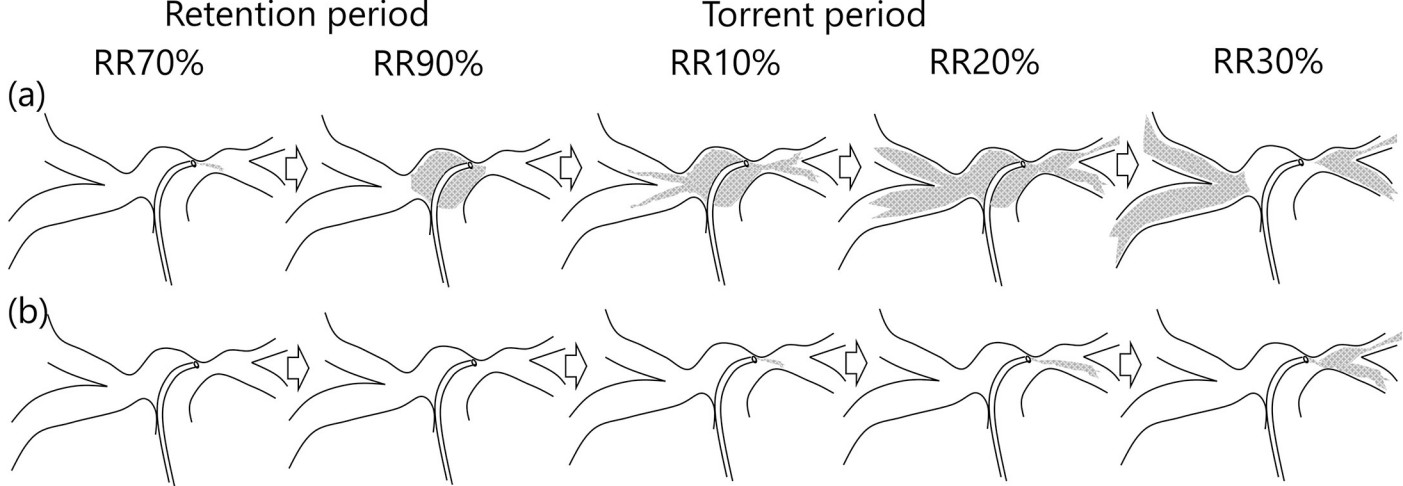

**Fig 5. Typical schematic demonstration of angiographic contrast agent injected into the pulmonary trunk through a catheter.** (a) If contrast agent is injected over 50–90% of the R-R interval in the retention period, the agent can be retained and well mixed during the retention period, then simultaneously discharged to both pulmonary branches over 10% of the R-R interval in the torrent period. (b) However, if the contrast is injected during 0–50% or 90–100% of the R-R interval in the retention period, the contrast agent flows preferentially to the unilateral pulmonary branch without retention or mixing.

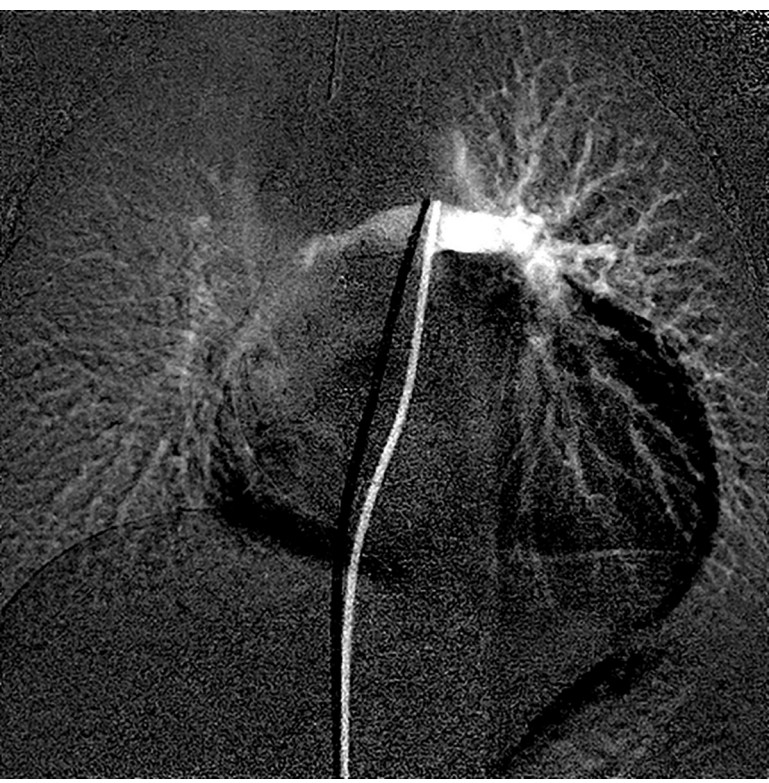

**Fig 6. Unilateral distribution of contrast agent injected into the main pulmonary trunk in the first cardiac cycle after injection.** Even if contrast agent is injected into the main pulmonary trunk, it is unilaterally distributed mainly to the left pulmonary artery during the first cardiac cycle after injection. Therefore, accurate measurement of the right-to-left pulmonary blood flow ratio with our model should be performed over 50–90% of the R-R interval during the retention period.

The net increase in signal intensity during the torrent period during the second cardiac cycle is calculated for each region: $S_{XA}^{Left}$ on the left (green arrow) and $S_{XA}^{Right}$ on the right (blue arrow) (**Fig 4**). The ratio of the increases in the signal intensity curves over the torrent period during the second cardiac cycle, $S_{XA}^{Right} : S_{XA}^{Left}$, determines the ratio of the right and left pulmonary blood flow.

Then, we investigated the feasibility and accuracy of our method in a pilot clinical setting with patients.

## Material and methods

Before testing our kinetic model on patients, we obtained approval of our protocol and study design by the Institutional Review Board and ethical committee of Nagano Children's Hospital (approval number IRB-28-1). After obtaining written informed consent from patients or their parents, 11 consecutive patients with congenital heart disease were enrolled in this pilot study and underwent XA and LS between September and November 2016.

The sample size of the patients in this pilot study was estimated by Fisher Z transformation with a confidence interval of 0.6 and a predicted confidence coefficient of >0.8. Patients whose pulmonary blood flow was supplied by multiple vessels, patients who had extra blood supply in addition to the main pulmonary artery, patients who had lacked imaging of the lung field, and patients who had overlapping images of the main pulmonary artery were excluded. Of the 11 initial patients, seven who met the inclusion criteria were analyzed.

## A. Acquisition

LS was performed using an e.cam with an e.soft workstation (Canon Medical Systems Corporation, Japan) using $^{99m}$Tc-MAA (radionuclide) as a radioisotope tracer. Planar images of both lungs in 6 directions, including the anterior-posterior (AP) and posterior-anterior (PA) directions, in a large ROI covering the entire field were acquired with a LEHR collimator, and the counts of each lung were averaged from both the AP and PA images. The counts were then converted to radioisotope tracer volumes using a predetermined calibration factor to obtain quantitative pulmonary blood flow.

X-ray pulmonary angiography was performed using a cardiovascular X-ray imaging system (Canon Medical Systems Corporation, Japan). The imaging parameters were as follows: field of view 5–8 inches, fixed tube voltage, pulse rate 30 frames per second, image matrix size 1024 by 1024, and no automatic brightness control or nonlinear image postprocessing. Total acquisition time was 6–10 seconds. Iodine contrast agent (Iopaque 300, Fuji Pharma, Japan) was injected as a bolus (1 ml/kg/second) in the pulmonary trunk through a 4–6 Fr catheter. The images were acquired continuously starting 1 second prior to contrast injection until all contrast agent was washed out from the lung field to the descending aorta on the AP projection. All images were stored in a workstation in DICOM format.

The examination interval between the XA and LS scans was 1–3 days.

## B. Image analysis

All image data were analyzed by in-house software using data processing protocol described in the previous section. The baseline image obtained prior to contrast injection was subtracted from subsequent, consecutive frames. The time window for calculation was set to the torrent period during the second cardiac cycle. The starting frame of the time window for calculation was set at the frame in which the contrast agent arrived at the first branch of the pulmonary artery, and the ending frame of the time window was set at the frame in which the contrast agent had filled the entire lung field. The net increase in signal intensity in the time window was analyzed in both the right and left ROI. The right-to-left ratio of pulmonary blood flow was obtained by the ratio of the net increase in signal intensity in each lung ROI.

For visual assessment, an image of the net increase in signal intensity in each pixel was generated. Furthermore, the generated image was blurred by a Gaussian filter followed by grayscale inversion. The blurred image was then visually and qualitatively compared with LS.

## C. Statistical analysis

Data analysis was performed using ImageJ (NIH, USA) and Microsoft Excel. Statistical analysis was performed using R version 3.3.0 (R Foundation for statistical computing, Vienna, Austria).

# Results

The demographic data of the enrolled patients are shown in Table 1. Of the 7 patients, three presented with complete transposition of the great artery (TGA) after an arterial switch operation (ASO), one had double outlet of the right ventricle (DORV) after pulmonary arterial banding (PAB), one had a complete atrioventricular septal defect: (cAVSD) after PAB, and two had a repair of a total anomalous pulmonary venous connection (TAPVC). The ages ranged from 6 to 71 months (mean 21.5 months), the body weight ranged from 7.2–17.1 kg (mean 8.6 kg), and the height ranged from 60 to 108 cm (mean 76 cm).

The ratios of the right-to-left pulmonary blood flow distribution obtained with each modality (XA, LS) are demonstrated in Table 2 and Fig 7. The scatterplot of the measurements from

**Table 1. Patient demographic data.**

| Patient | Age (years) | Sex | Body height (cm) | Body weight (kg) | Diagnosis |
|---|---|---|---|---|---|
| a | 1 | Male | 70 | 7.4 | TGA, s/p ASO |
| b | 5 | Female | 108 | 17.1 | TGA, s/p ASO |
| c | 1 | Male | 70 | 7.4 | TGA, s/p ASO, s/p PTA |
| d | 1 | Male | 78 | 7.3 | DORV, s/p PAB |
| e | 0 | Female | 60 | 4.7 | cAVSD, s/p PAB |
| f | 1 | Male | 78 | 9.4 | s/p TAPVC |
| g | 1 | Male | 71 | 7.2 | s/p TAPVC |

TGA, transposition of the great arteries; ASO, arterial switch operation; PTA, percutaneous angioplasty for left pulmonary artery; DORV, double outlet of the right ventricle; PAB, pulmonary artery banding; cAVSD, complete atrioventricular septal defect; TAPVC, total anomalous pulmonary venous connection; s/p, status post

XA and LS show a good linear correlation with a slope of 1.2 and a Pearson correlation coefficient of 0.91 (p<0.005).

The net increase in signal intensity by XA is shown in the left column of **Fig 8**, and the corresponding blurred images are shown in the middle column of **Fig 8**. Qualitatively, these images are in good agreement with the LS results, demonstrated in the right column of **Fig 8**.

Examples of the detailed results for one patient are shown in **Figs 9** and **10**. **Fig 9** shows consecutive XA images following subtraction of the baseline image in the time window for calculation. **Fig 10** is the time signal intensity curve of the patient.

## Discussion

### A. Major findings

Quantitative assessment of pulmonary blood flow distribution is important when determining the clinical indications for treating pulmonary arterial branch stenosis. LS is currently the gold standard for quantitative pulmonary blood flow measurement. However, LS is expensive, has difficulty in achieving real-time assessment, requires additional sedation, and exposes the patient to ionizing radiation. XA has been used routinely to assess the anatomy and qualitative blood flow distribution of the pulmonary arteries, especially in pediatric patients with congenital heart disease [15, 17, 18]. If a more detailed anatomy is required for diagnosis, such as the determination of stenosis or coarctation of the pulmonary artery, digital subtraction angiography can be used. However, routine XA cannot provide the quantitative blood flow distribution to each lung, which is usually needed to determine indications for further catheter or surgical intervention to local stenoses of the pulmonary artery.

**Table 2. Comparison of the right-to-left ratios between X-ray pulmonary angiography (XA) and lung perfusion scintigraphy (LS).**

| Patient | Number of frames in the time-window for calculation (XA) | Cardiac phase during the time-window for calculation (XA) | Right-to-left ratio using XA | Right-to-left ratio using LS |
|---|---|---|---|---|
| a | 5 frames | 13–43% R-R | 82:18 | 81:19 |
| b | 5 frames | 8–26% R-R | 61:39 | 59:41 |
| c | 5 frames | 11–41% R-R | 67:33 | 69:31 |
| d | 5 frames | 15–44% R-R | 30:70 | 43:57 |
| e | 5 frames | 10–34% R-R | 34:66 | 45:55 |
| f | 5 frames | 22–50% R-R | 64:36 | 55:45 |
| g | 9 frames | 92–41% R-R | 52:48 | 46:54 |

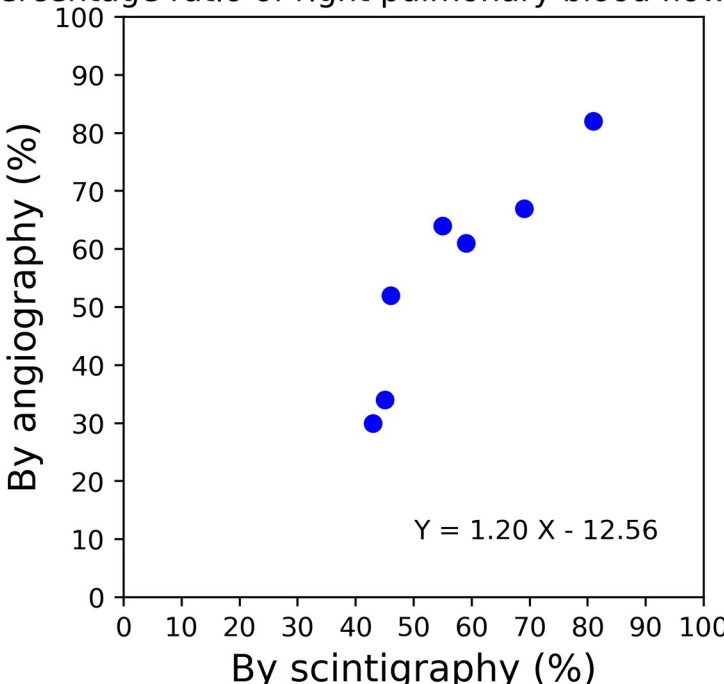

**Fig 7. Comparison of the right-to-left ratios between X-ray pulmonary angiography (XA) and lung perfusion scintigraphy (LS).** The values on each axis are the percentages of right pulmonary blood flow distribution relative to the total pulmonary blood flow. The percentage of right pulmonary blood flow calculated by angiography was in good agreement with that calculated by lung scintigraphy, with a slope of 1.2 and a Pearson correlation coefficient of 0.91 ($p < 0.005$).

In this research study, we developed a new approach that can measure the right-to-left ratio of the pulmonary blood flow distribution using conventional XA. The results of the new approach are in good agreement with those of LS qualitatively and quantitatively. Our novel approach presented in this paper is original and unique in the following aspects. Our method uses conventional XA without any further hardware modification for image acquisition and does not deviate from the standard of care workflow. The analysis is simple and straightforward with good reproducibility because the principal model is equivalent to LS, and quantitative assessment of blood flow is assured by calculating the net increase in signal intensity in the inflow phase, where blood stays in the pulmonary arteries and capillary bed. Our method is robust to subtle motions of the patient and deformations of the lung because the aim of our technology is to measure the right-to-left ratio of pulmonary blood flow; thus, arterial input function is not needed, and a large ROI that covers the entire lung field of each lung can be used.

In this regard, our method brings many benefits to clinical practice, especially in many aspects of pediatric cardiology. One can easily assess the pulmonary blood flow distribution only by routine pulmonary angiography without any further specialized equipment aside from this unique software. One can perform instant decision-making for treatment during the session in the catheterization theater without a long waiting period. Furthermore, our method could take the place of scintigraphy, which usually requires another session for the examination and additional radiation exposure with anesthesia. As a result, our approach could substantially reduce the burden on patients and medical care providers.

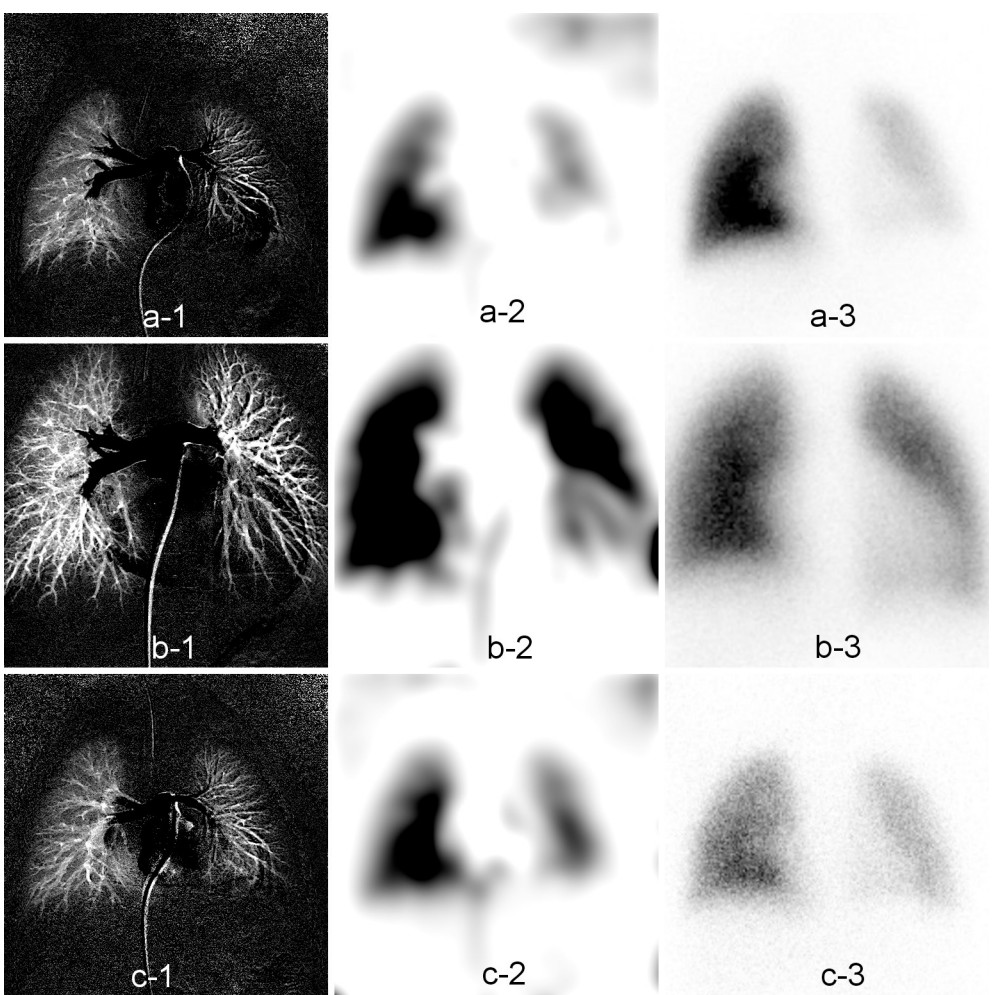

**Fig 8. Comparison between X-ray pulmonary angiography (XA) and lung perfusion scintigraphy (LS) images for three representative patients.** The net increase in signal intensity by XA is demonstrated in the left column (a-1, b-1, c-1). The images in the middle column are the XA images after grayscale inversion and blurring by Gaussian filtration (a-2, b-2, c-2). The images in the right column were obtained with lung scintigraphy (a-3, b-3, c-3).

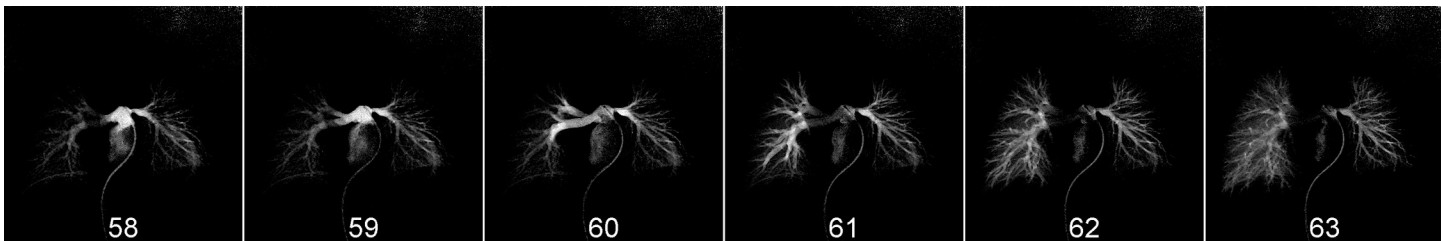

**Fig 9. Baseline subtracted images by X-ray pulmonary angiography (XA).** The numbers at the bottom of each image are the frame number of acquisition. The torrent period during the second cardiac cycle started from frame 58. Therefore, the time window for calculation was set to start from frame 59 (13% of the R-R interval in the torrent period), in which the contrast agent had flowed from the main pulmonary trunk to the first branch of both the right and left pulmonary artery. The time window for acquisition ended at frame 63 (43% of the R-R interval in the torrent period), in which the contrast agent had filled the entire lung field before passing through. The difference in the signal intensity between frames 59 and 63 of the ROI in each lung was calculated as the net increase in signal intensity.

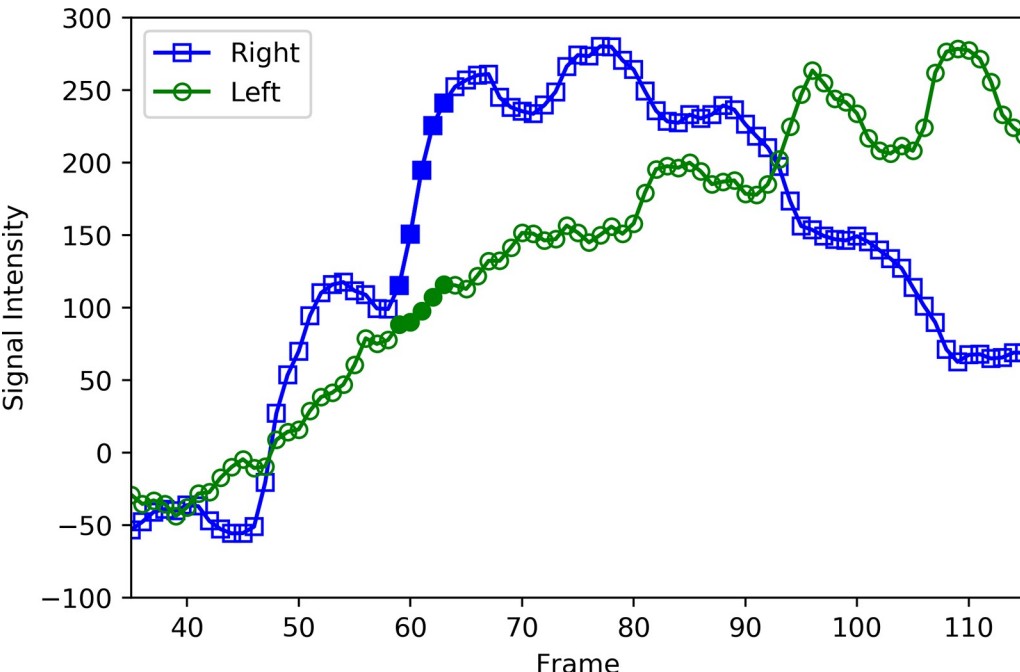

**Fig 10. Actual time-signal intensity curve calculated from the X-ray pulmonary angiography (XA) of the patient of Fig 9.** The signal intensity of the right lung field at each frame is plotted as an open blue square and that of the left lung as an open green circle. The closed squares and circles represent the signal intensities from frames 59 to 63, which were used for calculation. The net increase in signal intensity from frame 59 to 63 in the right lung was 126, while that in the left lung was 27. Hence, the ratio of pulmonary blood flow distribution was 126:27 (82:18).

## B. Limitation

Although our novel approach has been validated by a pilot cohort, the number of patients may not have been sufficient to elucidate its accuracy in real-world practice. A large-scale clinical study is necessary to validate our method and to demonstrate its clinical relevance so that it can be adopted in daily practice. In addition, technical limitations related to the size of the X-ray detector prevent coverage of the entire lung field in adult patients. For image acquisition, at least 30 frames per second is required for pediatric patients, but this may be reduced in adult patients. Faster frame rates are desirable to improve the accuracy and set an exact time window for the calculation of signal intensity by increasing the number of data points to be analyzed. Finally, automated setting of the ROI and time window for calculation may improve our workflow and avoid operator-dependent errors.

## Conclusions

A novel imaging technology for quantitatively assessing the right-to-left ratio of the pulmonary blood flow distribution by conventional X-ray pulmonary angiography was demonstrated. This system has great potential for clinical use given its accuracy, convenience, ease of operability and real-time nature. The results of the new approach are in good agreement with those of routine lung perfusion scintigraphy. Therefore, this novel imaging technology could serve as a substitute for lung perfusion scintigraphy. Such an approach would improve the clinical workflow and reduce the radiation exposure and burden to the patient. Further clinical studies are desired to translate the proposed method to a routine clinical workflow.

## Supporting information

**S1 Table. Time signal intensity data.**
(CSV)

## Acknowledgments

We thank Hiroyuki Obata (Radiologist, Nagano Children's Hospital) for his clinical setting of the angiography apparatus to obtain good images and Dr. Hatem Mehrez for his contribution to proofreading the English in the manuscript.

## Author Contributions

**Conceptualization:** Takuya Sakaguchi, Satoshi Yasukochi.

**Data curation:** Yuichiro Watanabe, Kohta Takei.

**Formal analysis:** Yuichiro Watanabe, Kohta Takei.

**Funding acquisition:** Masashi Hirose.

**Investigation:** Takuya Sakaguchi, Yuichiro Watanabe, Kohta Takei.

**Methodology:** Takuya Sakaguchi, Satoshi Yasukochi.

**Project administration:** Masashi Hirose, Satoshi Yasukochi.

**Resources:** Yuichiro Watanabe, Kohta Takei.

**Software:** Yuichiro Watanabe.

**Supervision:** Takuya Sakaguchi, Satoshi Yasukochi.

**Validation:** Takuya Sakaguchi, Satoshi Yasukochi.

**Visualization:** Takuya Sakaguchi, Yuichiro Watanabe.

**Writing – original draft:** Takuya Sakaguchi.

**Writing – review & editing:** Takuya Sakaguchi, Yuichiro Watanabe, Masashi Hirose, Kohta Takei, Satoshi Yasukochi.

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
