## [Decision Letter · Decision Letter 0]

29 Mar 2021

PONE-D-20-19770

A novel diagnostic approach for pulmonary blood flow distribution assessment using conventional X-ray angiography

PLOS ONE

Dear Dr. Satoshi Yasukochi,

Thank you for submitting your manuscript to PLOS ONE. After careful consideration, we feel that it has merit but does not fully meet PLOS ONE’s publication criteria as it currently stands. Therefore, we invite you to submit a revised version of the manuscript that addresses the points raised during the review process.

We look forward to receiving your revised manuscript.

Kind regards,

Alon Harris

Academic Editor

PLOS ONE

Journal Requirements:

3. You indicated that you had ethical approval for your study. In your Methods section, please ensure you have also stated whether you obtained consent from parents or guardians of the minors included in the study or whether the research ethics committee or IRB specifically waived the need for their consent.

4.  Thank you for including your ethics statement:

"The Institutional Review Board of Nagano Children’s Hospital approved all procedures. (IRI-28-1)".

i) Please amend your current ethics statement to confirm that your named institutional review board or ethics committee specifically approved this study.

ii) Once you have amended this/these statement(s) in the Methods section of the manuscript, please add the same text to the “Ethics Statement” field of the submission form (via “Edit Submission”).

5.  Thank you for stating the following in the Financial Disclosure section:

"This research was funded by Cannon Medical Systems Corporation."

We note that one or more of the authors have an affiliation to the commercial funders of this research study : Canon Medical Systems Corporation.

5.1. Please provide an amended Funding Statement declaring this commercial affiliation, as well as a statement regarding the Role of Funders in your study. If the funding organization did not play a role in the study design, data collection and analysis, decision to publish, or preparation of the manuscript and only provided financial support in the form of authors' salaries and/or research materials, please review your statements relating to the author contributions, and ensure you have specifically and accurately indicated the role(s) that these authors had in your study. You can update author roles in the Author Contributions section of the online submission form.

5.2. Please also provide an updated Competing Interests Statement declaring this commercial affiliation along with any other relevant declarations relating to employment, consultancy, patents, products in development, or marketed products, etc.  

Reviewers' comments:

Reviewer's Responses to Questions

**Comments to the Author**

1. Is the manuscript technically sound, and do the data support the conclusions?

Reviewer #1: Yes

Reviewer #2: Yes

2. Has the statistical analysis been performed appropriately and rigorously? 

Reviewer #1: N/A

Reviewer #2: N/A

3. Have the authors made all data underlying the findings in their manuscript fully available?

Reviewer #1: Yes

Reviewer #2: Yes

4. Is the manuscript presented in an intelligible fashion and written in standard English?

Reviewer #1: Yes

Reviewer #2: Yes

5. Review Comments to the Author

Reviewer #1: It is a new method and as stated it it is a preliminary study, to verify the theoretical assumption and calculation. Looking at this study at this viewpoint, it is interesting, but without large comparative study it has no clinical relevance, at the moment.

Reviewer #2: The authors presented an interesting pilot study evaluating the feasibility and accuracy of X-ray pulmonary angiography to measure pulmonary blood flow distribution in each lung compared to lung perfusion scintigraphy. The manuscript is with merit and the findings are worth reporting. However, before publication could be considered, the authors should revise the manuscript and address the following concerns.

- MATERIAL AND METHODS

o Participants: the authors should provide an indication of the criteria used to select the study patients

o Statistics: The author should provide some justification of the study n

- RESULTS

o The authors should indicate in the first paragraph of the results the characteristics of the study participants (demographics, age, gender, diagnosis etc)

Minor revisions

- Line 336: “Our method” should be replaced with “our method” (the letter “o” should not be capitalized after a colon)

- Figures/Tables: please revise all the legends and provide the complete explanation of the abbreviations used.

6. PLOS authors have the option to publish the peer review history of their article (what does this mean?). If published, this will include your full peer review and any attached files.

Reviewer #1: No

Reviewer #2: No

---

## [Author Response · Author response to Decision Letter 0]

24 Apr 2021

Response to Reviewers

We wish to thank the editors and reviewers for the constructive suggestions in relation to the submission of the original article on “A novel diagnostic approach for pulmonary blood flow distribution assessment using conventional X-ray angiography” (PONE-D-20-19770). We have revised the manuscript in accordance with your suggestions, and revisions are indicated in red in a file labeled ‘Revised Manuscript with Track Changes’.

In this revised manuscript, we increased the number of patients to demonstrate the clinical relevance of our new method of measuring pulmonary blood flow distribution in each lung using X-ray pulmonary angiography. 

PONE-D-20-19770

A novel diagnostic approach for pulmonary blood flow distribution assessment using conventional X-ray angiography

PLOS ONE

Journal Requirements:

REPLY: We have amended manuscript to meet PLOS ONE’s style requirements. 

REPLY: Our manuscript have been copyedited by our North American colleague named Dr. Hatem Mehrez 

3. You indicated that you had ethical approval for your study. In your Methods section, please ensure you have also stated whether you obtained consent from parents or guardians of the minors included in the study or whether the research ethics committee or IRB specifically waived the need for their consent.

REPLY: We had obtained written informed consent from patients’ parents. We have added a statement about obtaining consent in the Material and Methods section. 

4. Thank you for including your ethics statement:

"The Institutional Review Board of Nagano Children’s Hospital approved all procedures. (IRI-28-1)".

i) Please amend your current ethics statement to confirm that your named institutional review board or ethics committee specifically approved this study.

ii) Once you have amended this/these statement(s) in the Methods section of the manuscript, please add the same text to the “Ethics Statement” field of the submission form (via “Edit Submission”).

REPLY: We have amended ethics statement in the Material and Methods section. 

5. Thank you for stating the following in the Financial Disclosure section:

"This research was funded by Cannon Medical Systems Corporation."

We note that one or more of the authors have an affiliation to the commercial funders of this research study : Canon Medical Systems Corporation.

5.1. Please provide an amended Funding Statement declaring this commercial affiliation, as well as a statement regarding the Role of Funders in your study. If the funding organization did not play a role in the study design, data collection and analysis, decision to publish, or preparation of the manuscript and only provided financial support in the form of authors' salaries and/or research materials, please review your statements relating to the author contributions, and ensure you have specifically and accurately indicated the role(s) that these authors had in your study. You can update author roles in the Author Contributions section of the online submission form.

REPLY: We have added Funding section and declare commercial affiliation as the editor showed. 

5.2. Please also provide an updated Competing Interests Statement declaring this commercial affiliation along with any other relevant declarations relating to employment, consultancy, patents, products in development, or marketed products, etc. 

REPLY: We have added Competing Interests Statement also in Funding section. 

Reviewers' comments:

Reviewer's Responses to Questions

Comments to the Author

1. Is the manuscript technically sound, and do the data support the conclusions?

Reviewer #1: Yes

Reviewer #2: Yes

2. Has the statistical analysis been performed appropriately and rigorously?

Reviewer #1: N/A

Reviewer #2: N/A

3. Have the authors made all data underlying the findings in their manuscript fully available?

Reviewer #1: Yes

Reviewer #2: Yes

4. Is the manuscript presented in an intelligible fashion and written in standard English?

Reviewer #1: Yes

Reviewer #2: Yes

5. Review Comments to the Author

Reviewer #1: It is a new method and as stated it it is a preliminary study, to verify the theoretical assumption and calculation. Looking at this study at this viewpoint, it is interesting, but without large comparative study it has no clinical relevance, at the moment.

REPLY: In the first manuscript, we had conducted the study as a pilot study. However, we have increased the number of patients from 3 to 7 to demonstrate clinical relevance and to meet the statistically required sample size in this revised manuscript. 

Reviewer #2: The authors presented an interesting pilot study evaluating the feasibility and accuracy of X-ray pulmonary angiography to measure pulmonary blood flow distribution in each lung compared to lung perfusion scintigraphy. The manuscript is with merit and the findings are worth reporting. However, before publication could be considered, the authors should revise the manuscript and address the following concerns.

- MATERIAL AND METHODS

o Participants: the authors should provide an indication of the criteria used to select the study patients

REPLY: We have added inclusion and exclusion criteria for this study in Material and Methods section. 

o Statistics: The author should provide some justification of the study n

REPLY: We have added a sentence to show the statistical justification of the sample size in Material and Methods section. 

- RESULTS

o The authors should indicate in the first paragraph of the results the characteristics of the study participants (demographics, age, gender, diagnosis etc)

REPLY: We have added Table 1 which represents demographic data of the enrolled patients. 

Minor revisions

- Line 336: “Our method” should be replaced with “our method” (the letter “o” should not be capitalized after a colon)

REPLY: We have uncapitalized the word “Our”. 

- Figures/Tables: please revise all the legends and provide the complete explanation of the abbreviations used.

REPLY: We have added explanations of the abbreviations used in figures and tables. 

6. PLOS authors have the option to publish the peer review history of their article (what does this mean?). If published, this will include your full peer review and any attached files.

Do you want your identity to be public for this peer review? For information about this choice, including consent withdrawal, please see our Privacy Policy.

Reviewer #1: No

Reviewer #2: No

---

## [Decision Letter · Decision Letter 1]

9 Jun 2021

A novel diagnostic approach for assessing pulmonary blood flow distribution using conventional X-ray angiography

PONE-D-20-19770R1

Dear Dr. Satoshi Yasukochi,

We’re pleased to inform you that your manuscript has been judged scientifically suitable for publication and will be formally accepted for publication once it meets all outstanding technical requirements.

Kind regards,

Alon Harris

Academic Editor

PLOS ONE

Additional Editor Comments (optional):

Reviewers' comments:

Reviewer's Responses to Questions

**Comments to the Author**

1. If the authors have adequately addressed your comments raised in a previous round of review and you feel that this manuscript is now acceptable for publication, you may indicate that here to bypass the “Comments to the Author” section, enter your conflict of interest statement in the “Confidential to Editor” section, and submit your "Accept" recommendation.

Reviewer #1: All comments have been addressed

Reviewer #2: All comments have been addressed

2. Is the manuscript technically sound, and do the data support the conclusions?

Reviewer #1: Yes

Reviewer #2: Yes

3. Has the statistical analysis been performed appropriately and rigorously? 

Reviewer #1: N/A

Reviewer #2: Yes

4. Have the authors made all data underlying the findings in their manuscript fully available?

Reviewer #1: Yes

Reviewer #2: Yes

5. Is the manuscript presented in an intelligible fashion and written in standard English?

Reviewer #1: Yes

Reviewer #2: Yes

6. Review Comments to the Author

Reviewer #1: The authors addressed and corrected all the points of the reviewers. The study can be accepted and published.

Reviewer #2: The authors addressed all the comments of the reviewers and the manuscript can be now accepted for publication.

7. PLOS authors have the option to publish the peer review history of their article (what does this mean?). If published, this will include your full peer review and any attached files.

Reviewer #1: No

Reviewer #2: No

---

## [Editor Report · Acceptance letter]

15 Jun 2021

PONE-D-20-19770R1 

A novel diagnostic approach for assessing pulmonary blood flow distribution using conventional X-ray angiography 

Dear Dr. Yasukochi:

I'm pleased to inform you that your manuscript has been deemed suitable for publication in PLOS ONE. Congratulations! Your manuscript is now with our production department. 

Kind regards, 

on behalf of

Dr. Alon Harris 

Academic Editor

PLOS ONE